# Predictive Factors of Piperacillin Exposure and the Impact on Target Attainment after Continuous Infusion Administration to Critically Ill Patients

**DOI:** 10.3390/antibiotics12030531

**Published:** 2023-03-07

**Authors:** Javier Martínez-Casanova, Erika Esteve-Pitarch, Helena Colom-Codina, Víctor Daniel Gumucio-Sanguino, Sara Cobo-Sacristán, Evelyn Shaw, Kristel Maisterra-Santos, Joan Sabater-Riera, Xosé L. Pérez-Fernandez, Raül Rigo-Bonnin, Fe Tubau-Quintano, Jordi Carratalà, Ariadna Padullés-Zamora

**Affiliations:** 1Pharmacy Department, Hospital Universitari de Bellvitge, 08907 Hospitalet de Llobregat, Spain; 2Farmacoteràpia, Farmacogenètica i Tecnologia Farmacèutica, Institut d’Investigació Biomèdica de Bellvitge, IDIBELL, 08907 Hospitalet de Llobregat, Spain; 3Pharmacy Department, Hospital Universitari de Tarragona Joan XXIII, 43005 Tarragona, Spain; 4Pharmacy and Pharmaceutical Technology and Physical Chemistry Department, Universitat de Barcelona, 08028 Barcelona, Spain; 5Critical Care Department, Hospital Universitari de Bellvitge, 08907 Hospitalet de Llobregat, Spain; 6Infectious Diseases Department, Hospital Universitari de Bellvitge, 08907 Hospitalet de Llobregat, Spain; 7Epidemiologia de les Infeccions Bacterianes, Patologia Infecciosa i Transplantament, Institut d’Investigacio Biomedica de Bellvitge, IDIBELL, 08907 Hospitalet de Llobregat, Spain; 8Centro de Investigación Biomédica en Red de Enfermedades Infecciosas (CIBERINFEC), Instituto de Salud Carlos III, 28029 Madrid, Spain; 9Clinical Laboratory Department, Hospital Universitari de Bellvitge, 08907 Hospitalet de Llobregat, Spain; 10Department of Microbiology, Hospital Universitari de Bellvitge-IDIBELL, 08907 Hospitalet de Llobregat, Spain; 11Centro de Investigación Biomédica en Red de Enfermedades Respiratorias (CIBERES), Instituto de Salud Carlos III, 28029 Madrid, Spain; 12Campus Ciencias de la Salud de Bellvitge, University of Barcelona, 08907 Hospitalet de Llobregat, Spain

**Keywords:** beta-lactam antibiotic, piperacillin, critically ill, continuous infusion, creatinine clearance, population pharmacokinetic approach, therapeutic drug monitoring

## Abstract

Critically ill patients undergo significant pathophysiological changes that affect antibiotic pharmacokinetics. Piperacillin/tazobactam administered by continuous infusion (CI) improves pharmacokinetic/pharmacodynamic (PK/PD) target attainment. This study aimed to characterize piperacillin PK after CI administration of piperacillin/tazobactam in critically ill adult patients with preserved renal function and to determine the empirical optimal dosing regimen. A total of 218 piperacillin concentrations from 106 patients were simultaneously analyzed through the population PK approach. A two-compartment linear model best described the data. Creatinine clearance (CL_CR_) estimated by CKD-EPI was the covariate, the most predictive factor of piperacillin clearance (CL) interindividual variability. The mean (relative standard error) parameter estimates for the final model were: CL: 12.0 L/h (6.03%); central and peripheral compartment distribution volumes: 20.7 L (8.94%) and 62.4 L (50.80%), respectively; intercompartmental clearance: 4.8 L/h (26.4%). For the PK/PD target of 100% *f*T_>1×MIC_, 12 g of piperacillin provide a probability of target attainment > 90% for MIC < 16 mg/L, regardless of CL_CR_, but higher doses are needed for MIC = 16 mg/L when CL_CR_ > 100 mL/min. For 100% *f*T_>4×MIC_, the highest dose (24 g/24 h) was not sufficient to ensure adequate exposure, except for MICs of 1 and 4 mg/L. Our model can be used as a support tool for initial dose guidance and during therapeutic drug monitoring.

## 1. Introduction

Sepsis in intensive care units (ICUs) is a major health problem worldwide due to the high morbidity and mortality associated with it [1,2,3]. An early and appropriate antimicrobial therapy is a key component for optimizing the chances of survival, with a focus on timely administration, the appropriateness of the antimicrobial spectrum and dosing optimization from the beginning of the treatment [1].

Piperacillin/tazobactam (P/T) is a β-lactam/β-lactamase inhibitor that exhibits activity against a broad range of microorganisms, including gram-positive and gram-negative aerobic and anaerobic bacteria. It is routinely prescribed for the empirical and targeted treatment of infections in critically ill patients with sepsis or septic shock [4]. Its antibacterial activity is time-dependent, so its efficacy is related to the time (T) during which the free drug concentration (*f*) is maintained above the minimum inhibitory concentration (MIC) (%*f*T_>MIC_) [5]. The pharmacokinetic/pharmacodynamic (PK/PD) target is based on this premise, and despite the lack of consensus as to which is optimal, for critically ill patients, treatment is considered adequate when free concentrations of the antibiotic are at least 100% *f*T_>4×MIC_ [6,7].

Critically ill patients present pathophysiological changes, such as third spacing (fluid extravasation into interstitial space), increased cardiac output, hypoalbuminemia or elevated creatinine clearance that may affect antibiotic pharmacokinetics (PKs), leading to sub-therapeutic antibiotic concentrations [4,8,9,10]. Moreover, the presence of chronic comorbidity can further exacerbate the pathophysiological changes commonly encountered during critical illness [10]. It is worthy of note that such alterations are multifactorial and may not be present in all patients, further decreasing the predictability of PK/PD in critically ill patients.

P/T, as the rest of β-lactam antibiotics, is a hydrophilic antibiotic with moderate protein binding (30%) and mainly cleared by renal excretion (i.e., glomerular filtration and active tubular secretion). These characteristics may cause its PK to be particularly affected by the above-mentioned events [11]. Previous studies have found that administering P/T as extended or continuous infusion (CI) rather than intermittent infusion (II) might improve PK/PD target attainment in these situations [1,11,12,13,14,15]. In fact, the administration of P/T by CI preceded by a loading dose is nowadays a widespread practice in the ICU.

Population PK (PPK) models are powerful descriptive and predictive support tools for the implementation of therapeutic drug monitoring (TDM). While several piperacillin PPK models have been previously published in critically ill adult patients, most of them were the target population undergoing renal replacement therapy (RRT) and are, therefore, not applicable to other population groups [16,17,18,19,20]. Furthermore, in models developed in patients without RRT, CL_CR_ was frequently identified as the strongest covariate with a significant linear relationship to piperacillin clearance [21,22,23,24,25,26,27,28]. However, in the vast majority, P/T administered as II was used to predict PK behavior in CI [22,25,26,27,28]. Caution should be taken when making dose estimates in CI based on models developed in II, as non-linear PKs have been postulated with the latter form of administration due to saturation of piperacillin elimination [29,30]. This phenomenon could make such estimates not fully equivalent since CI-predicted concentrations could be overestimated, leading to underexposure. The development of PPK models based on CI data is, therefore, necessary.

In a previous study by our group [31], statistical analyses showed that standard total daily doses of piperacillin administered by CI were not enough to achieve optimal PK/PD targets when the minimum inhibitory concentration (MIC) was above 8 mg/L. In addition, elevated creatinine clearance (CL_CR_), followed by neurocritical status and mechanical ventilation, were identified as risk factors associated with subtherapeutic exposure. Therefore, a modeling approach is required to confirm previous results and to enable P/T dose optimization in the target population.

The aims of this study were: (i) to characterize the PK of piperacillin after administration of P/T as a CI in a population of critically ill adult patients with preserved renal function, using a population-based approach; and (ii) to determine the empirical optimal dosing regimen in this population.

## 2. Results

### 2.1. Patients and Datasets

A total of 218 samples from 106 patients were available for analysis. Patients’ demographic and clinical characteristics at baseline are reported in Table 1. The patients included were mainly men (67%), the median age was 65 years (interquartile range (IQR) 50–72) and the median CL_CR_ according to the Chronic Kidney Disease Epidemiology Collaboration formula (CL_CR_CKD-EPI) was 97 mL/min/1.73 m^2^ (IQR 86–114). Augmented renal clearance (ARC) was identified in 11 patients (10.38%). BMI values were indicative of non-obesity.

All patients underwent therapeutic drug monitoring based on the piperacillin measured concentrations, and only on four occasions from three different patients was the dose modified. The piperacillin plasma concentration-time profiles are displayed in Figure 1. The average number of samples per patient was 2.1. The peak and steady-state concentration (Cmax and Css) geometric mean values were 112 mg/L (IQR 69–184) and 39.5 mg/L (27–58), respectively.

### 2.2. Population PK Modeling

A two-compartment model with first-order elimination from the central compartment was considered to best describe piperacillin plasma concentration-time profiles. Between-patient variability (BPV) could only be associated with plasma clearance. Residual variability was best described by a proportional error model.

Inclusion of all the candidate covariates in the model, one at each step, identified renal function and administration of vasoactive drugs as the only covariates with a statistically significant effect on piperacillin plasma clearance (CL). CL_CR_ given by CKD-EPI resulted in the highest decrease in the minimum objective function value (MOFV) (∆MOFV = *−*29.88, *p* < 0.001), followed by the Cockcroft and Gault formula (CG) (∆MOFV = *−*27.27, *p* < 0.001) and then by the Modification of Diet in Renal Disease formula (MDRD-4) (∆MOFV = *−*18.62, *p* < 0.001). Between-patient variabilities associated with CL were reduced by 24.90% (CKD-EPI), 24.51% (CG) and 12.25% (MDRD-4). Although vasoactive drugs statistically reduced the MOFV (∆MOFV = *−*8.83, BPV = 10.67%), the sequential inclusion of the administration of vasoactive drugs on CL_CR_-models did not result in a significant decrease of the OFV for any of the CL_CR_ models considered.

According to these results, the final model only retained renal function as a covariate of CL, with a linear relationship between the typical value of plasma clearance and CL_CR_CKDEPI centered at its median value in the target population.

The parameter estimates of the base and final models as the bootstrap results are summarized in Table 2. All the PK parameters were estimated with adequate precision. The mean population values of all the model parameters were within the 95% confidence intervals estimated by the bootstrap method. The relative deviation between the true population value and the median value provided by bootstrap was lower than 10% for all the PK parameters. The condition number of the model was 4.40, suggesting no notable collinearity. Acceptable shrinkage values associated with ω^2^_CL_ and σ^2^ were found.

### 2.3. Model Evaluation

Goodness-of-fit plots of the final model are displayed in Figure 2. A random distribution around the identity line was defined for observed concentrations (DVs) versus population predicted values (PREDs) and DV versus individual predicted values (IPREDs). Conditional weighted residuals (CWRESs) versus time and individual weighted residuals (IWRESs) versus time also showed a random distribution around zero. All these plots were indicative of no model mis-specification (DV/PRED and CWRES/time) and adequate description of between-patient variability (DV/IPRED) and residual error (IWRES/time).

The visual inspection of the prediction-corrected, visual predictive checks (Figure 3) proved the descriptive and predictive capability of the model. The model adequately described the mean trend of the data. Overall, 50%, 2.5% and 97.5% of the observed data fell within the 95% confidence intervals of the corresponding percentiles of the simulated data, and most of the observed concentrations also fell within the 95% prediction interval.

### 2.4. Probability of Target Attainment and Cumulative Fraction of Response

Figure 4 shows the probability of target attainment (PTA) for each one of the PK/PD evaluated targets (100% *f*T_>1×MIC_ and 100% *f*T_>4×MIC_) for each piperacillin dosage regimen (8, 12, 16, 20 and 24 g given daily as CI), renal function cut-off (from 60 to 200 mL/min/1.73 m^2^ in steps of 20 mL/min/1.73 m^2^) and MIC scenarios (from 1 to 16 mg/L).

Cumulative fraction of response (CFR) for susceptible Pseudomonas aeruginosa isolates (MIC ≤ 16 mg/L) for the different piperacillin CI dosage regimens evaluated (preceded by a 4 g piperacillin loading dose) and for each CL_CR_CKD-EPI cut-off, considering the 2 PK/PD targets (100% *f*T_>1×MIC_ and 100% *f*T_>4×MIC_), are summarized in Table 3.

## 3. Discussion

This is the largest study (N = 106 patients) on predictive factors of piperacillin exposure and PK/PD target attainment in critically ill patients with normal renal function after CI administration of P/T. While previous studies on critically ill patients without RRT exist, some were mainly focused on the comparison among different types of administration (II or CI) [11,21]. In other studies, the impact of P/T administration as CI on PK/PD target attainment was investigated, but the target population consisted of patients with a wider range of variation in renal function (from around <30 to >130 mL/min) than in the current study [23,24].

Our large sample size has allowed us to better investigate the influence of new covariates other than those classically explored, such as renal function or weight. Indeed, the influences of neurocritical status, mechanical ventilation or drainage carriage were also evaluated. Moreover, unlike previous studies, the performance of the CKD-EPI formula, compared to others traditionally used (CG and MDRD-4), was assessed.

Although a sparse sample design was applied, our data allowed us to describe the piperacillin PK profile by a two-compartment disposition model with linear elimination. Probably, the sampling design, in which concentrations were measured after the loading dose, benefited the characterization of drug distribution. Hence, the current model resulted in a more adequate description of piperacillin concentrations than the previous one-compartment models reported after CI administration [23,24].

The finding of the first-order elimination agreed with previous studies and confirmed that no saturation of the piperacillin elimination process mediated through tubular secretion takes place during CI administration [23,24]. It should be noted that it has been postulated that saturation could also occur at therapeutic doses during CI administration [21]; nevertheless, a Michaelis–Menten or a parallel first-order/Michaelis–Menten elimination were tested and did not improve the data fit. This was in agreement with other studies that reported that saturable elimination should not be expected to be of major clinical importance in patients given daily doses from 6 to 18 g piperacillin [32].

In line with previous studies, CL_CR_ was identified as the most powerful predictor of piperacillin CL variability, despite the narrower renal function variation of our population [60–180 mL/min/1.73 m^2^] compared to theirs [23,24]. This finding is supported by the high contribution of renal excretion (70%) to the piperacillin elimination [33]. Due to the controversy as to which formula is the most suitable for predicting renal clearance in critically ill patients, the three most widely accepted (CKD-EPI, CG, MDRD-4) were evaluated in our study [34]. All of them significantly accounted for the variability of CL; however, CKD-EPI provided the best exposure predictions, followed by MDRD-4 and then by CG, confirming the findings of other authors with other drugs also renally excreted [35]. The fact that the IDMS-traceable creatinine was not used for the standardization of creatinine concentrations when the CG module was developed, and that its equation includes body weight instead of body surface area, could be reasons for the overestimation of renal function using this formula. Dhaese et al. also showed that MDRD-4 was superior to CG, but no comparison could be performed with CKD-EPI, as this was the first study in which this equation has been used to predict piperacillin exposure [23].

No influence of any of the other variables on piperacillin concentrations was demonstrated. Mechanical ventilation was able to decrease CL variability, but this was not considered significant. Further larger studies would be necessary to evaluate this association.

The piperacillin CL in our study (12 L/h) was higher than that reported by Dhaese et al. (8.38 L/h) and Klastrup et al. (6.43 L/h) [23,24]. This difference may be explained by the fact that they included patients with renal dysfunction, whereas in our case, in order to obtain a more accurate characterization of piperacillin PK in the subset of patients with preserved renal function, only patients with CL_CR_ ≥ 60 mL/min/1.73 m^2^ were included. These findings are supported by the results of our previous statistical analysis of the same data [31].

Differences were also encountered between distribution volumes, among studies. The volume of distribution (Vd) of piperacillin was significantly larger in our cohort of patients (83.1 L) than that estimated in two other previous studies (25.54 L and 35.8 L) [23,24]. Furthermore, the central compartment distribution volume (Vc) in our study was larger (20.7 L) than the one found by Roberts et al. in 16 patients treated either with CI or intermittent bolus (7.2 L) [11]. In our analysis, the Cmax values of 53 patients (50%) were included in the model. Considering that Vc estimation relies on Cmax values, the final Vc estimate by our model should be considered reliable. Differences in body weight among studies might also explain these discrepancies. However, neither body weight nor any other size or body composition metrics were statistically significant in any of the parameters of the model. The inclusion of body weight would have allowed a more accurate estimation of PK parameters and exposure predictions; therefore, further studies will be required to investigate the influence of this covariate. On the other hand, several factors lead to an increased Vd in the critically ill patient. The expansion of the interstitial space through capillary leakage, in addition to the intravenous fluid load suffered by these patients, results in a significant increase of the Vd of hydrophilic drugs and thus to a lower antibiotic concentration. This phenomenon is further aggravated by the occurrence of hypoalbuminemia, the presence of which should be considered in our cohort of patients (median albumin concentrations 29 g/L) [36,37].

Conflicting findings have been reported so as to put forward the clinical advantages that P/T CI can provide over other forms of administration [12,13,15,38]. However, a meta-analysis from randomized trials in patients with severe sepsis demonstrated a lower mortality rate in patients treated with beta-lactam by CI with respect to those treated by II [39]. In the same line, two recent meta-analyses comparing the administration of beta-lactams, including P/T, by prolonged versus II demonstrated a reduction in the clinical cure rate and mortality [40,41]. Moreover, there is growing evidence supporting beta-lactam CI over conventional II regimens for a greater achievement of the PK/PD target [11,12,13,14,15]. Thus, considering the need for rapid and adequate antibiotic coverage in the critically ill patient and the absence of adverse effects related to this form of administration, CI of beta-lactams preceded by an initial loading dose is currently a formal recommendation [1].

Since the internal validation endorsed the predictive capability of the model, the impact of elevated renal function and decreasing bacterial susceptibility on piperacillin PK/PD attainment when administered by CI was evaluated. Our simulations indicate that for the target of 100% *f*T_>1×MIC_, PTA was greater than 90% when using daily doses of 12 g CI for MICs of 1, 4 and 8 mg/L, regardless of renal function. However, when considering the worst case scenario (MIC of 16 mg/L), standard dosing regimens by CI (12 g/24 h and 16 g/24 h) would be not enough in patients with CL_CR_ greater than 100 and 140 mL/min/1.73 m^2^, respectively. Only the highest daily doses would provide adequate exposure for patients with ARC, which is in line with previous studies published [23,24]. In the rest of the scenarios, the achievement of a PTA above the cut-off point was closely dependent on renal function. When seeking the higher target *f*T_>4×MIC_ and for a MIC of 16 mg/L or susceptible *Pseudomonas aeruginosa* isolates (MIC ≤ 16 mg/L), even the highest dose is far from reaching the cut-off point off 90%. These results confirm those previously found by Dhaese et al., who concluded that the 24 g/24 h dose was insufficient to ensure adequate exposure in patients with CL_CR_ > 90 mL/min/1.73 m^2^ [23].

The present results highlight the influence of CL_CR_ on piperacillin CL and point out the great difficulty in achieving adequate treatment for critically ill patients without renal impairment when a more demanding PK/PD target is sought. The lack of consensus regarding at what the PK/PD index of beta-lactams in the critically ill patient should be established makes it difficult to provide solid dosing strategy recommendations. In fact, neither the %*f*T nor the threshold value to be reached are clearly defined, so that the range of targets fixed in the available studies may vary from 20% *f*T_>1×MIC_ to 100% *f*T_>5×MIC_ [6]. Available data suggest that maximum bactericidal activity occurs when the dose is maintained between four and five times the MIC. On this basis, beta-lactam treatment of critically ill patients is currently considered optimal when concentrations are at least 100% *f*T_>4×MIC_ [6,7]. Aware of the difficulties involved in achieving such a demanding objective, we decided to also evaluate the achievement of the 100% *f*T_>1×MIC_ target, which seems more realistic under certain conditions and has also proven to be effective. In any case, the simulations obtained are consistent with the fact that augmented renal clearance (ARC) is recognized as an important predictor of beta-lactam underexposure [36,42,43]. ARC is being increasingly reported in critically ill patients, and, although its impact on clinical outcome is unclear, it should be identified since patients may benefit from higher antimicrobial doses at the beginning of the antimicrobial treatment [36,43,44,45].

Patients were treated with P/T, but only piperacillin concentrations were measured. There is enough evidence supporting the existence of a high correlation between piperacillin and tazobactam concentrations [28,46,47]. In fact, a recent PK study showed that tazobactam concentrations were above the threshold in 92% of cases in which the piperacillin PK/PD target was achieved [47]. The similarity in the PK of these two compounds means that the determination of tazobactam is often considered dispensable, and consequently, studies in which it is evaluated are scarce. In previous PPK studies carried out in patients undergoing RRT [16,19], simulated tazobactam concentrations were found to be above threshold at all times. In a more recent study in which P/T was administered by II, tazobactam concentrations were predicted in critically ill patients with creatinine clearance up to 200 mL/min [28]. The tazobactam target was reached in 94% and 99% of the patients when doses of 1.5 g/24 h and ≥2 g/24 h were administered, thus P/T efficacy was considered to be uniquely related to piperacillin concentration. Although Wallemburg et al. considered the development of a joint model to adequately predict piperacillin concentrations [28], previous studies showed there was no interaction between them, and the PK/PD analysis was therefore based solely on the simulations performed with the piperacillin model [16,19]. Additional studies assessing tazobactam concentrations are needed to confirm the existence of such an association.

Other limitations may apply to our study. Only total drug concentrations were measured, while pharmacological activity and PK/PD targets are related to free fraction. Since the united fraction is considered to be 30% of the total, the free concentration was estimated [48]. Further, the study explores the likelihood of target attainment using different levels of bacterial susceptibility to optimize empirical dosing regimen, so no conclusions can be drawn regarding the appropriate drug exposure for any specific patient, for which TDM remains critical.

## 4. Material and Methods

### 4.1. Study Design

This was an observational prospective study conducted at the ICU of the Bellvitge University Hospital in Barcelona, Spain, during a 4-year period (August 2015–June 2019). Ethical approval was obtained from the local Ethics Committee (SFB-ATB-2014-01) and conducted in accordance with the Declaration of Helsinki. Written informed consent was requested from the patient or the closest relative before inclusion.

### 4.2. Patient Population

Critically ill patients under treatment with P/T administered as CI and with a preserved renal function (CL_CR_CKD-EPI ≥ 60 mL/min/1.73 m^2^) were eligible for the study. Patients receiving renal replacement therapy or extracorporeal membrane oxygenation, pregnant women and patients under 18 years were excluded.

### 4.3. Dosage, Sampling and Data Collection

Patients received an initial loading dose of P/T 4/0.5 g, administered in 30 min on day 1, followed by a continuous 24 h infusion of piperacillin (500 mg/h), i.e., 12 g piperacillin/1.5 g tazobactam in 150 mL 0.9% sodium chloride (80 mg/mL, stability of 24 h at 25 °C, 1 infusion/day). Patients who had already been on intermittent antibiotic treatment for at least 24 h did not receive the loading dose. This dose was subsequently modified, depending on the plasma concentrations achieved. Two different sampling periods were considered: shortly after the loading dose (30–60 min) and once steady-state conditions were attained (from 6 h after initiation of CI), from which Cmax and Css were measured, respectively.

The patients’ demographic (age, gender, weight, heigh, body mass index (BMI) and race) and clinical data (creatinine, urea, albumin, neurocritical status, mechanical ventilation, treatment with vasoactive drugs, and drainage carriage) were recorded at baseline and on each sampling occasion. CL_CR_ was calculated using three different equations: CKD-EPI, CG and MDRD-4. ARC, defined as CL_CR_CKD-EPI ≥ 130 mL/min/1.73 m^2^, was also assessed. The presence of neurological damage, use of mechanical ventilation, use of vasoactive drugs and presence of drainage were also prospectively recorded.

### 4.4. Bioanalysis

For piperacillin determination, approximately 3 mL of blood were collected in lithium-heparin tubes (Vacuette, Kremsmünster, Austria) and immediately refrigerated at 2–8 °C for a maximum of 30 min. Samples were then centrifuged at 2000× *g* for 10 min at (4 ± 1) °C, aliquoted and stored at (−75 ± 3) °C until analysis.

We analyzed total plasma concentrations of piperacillin using a previously validated method of ultra-performance liquid chromatography-tandem coupled to mass spectrometry (UHPLC-MS/MS) [49]. Briefly, the inter-day lower limit of quantification (LLOQ) was 0.54 mg/L (S/N ratio of 5.6), and the calibration curve ranged from 0.54 to 175 mg/L (a linear regression curve with a weighting scheme of 1/X).

### 4.5. Population Pharmacokinetics Analysis

#### 4.5.1. Base Model Development

A simultaneous analysis of all the piperacillin concentration-time data was performed with the PPK approach by means of nonlinear mixed-effects models implemented in NONMEM software, version 7.4 (ICON Development Solutions, Ellicott City, MD, USA) [50]. The first order conditional estimation method with interaction (FOCEI) was used throughout the model-building process. Graphical diagnostics were guided using Xpose version 4.2.1 implemented in the R version 3.3.2 and Perl speaks-NONMEM Toolkit (PsN) version 4.7.0 [51,52]. The Phoenix-WinNonlin version 64 8.2.0.4383 (Certara L.P., 1998–2018) was used for non-compartmental analyses of the data [53].

Models of one and two compartments were fitted to the concentration-time data. The models were parameterized in terms of apparent Vd, distributional clearance (CL_D_) and CL. First-order, Michaelis–Menten and parallel first-order/Michaelis–Menten kinetics were tested to describe the piperacillin elimination. BPV was evaluated for each PK parameter and modeled exponentially, assuming a log-normal distribution. Additive, proportional and combined (additive + proportional) models were compared to describe the residual error (RE) associated with drug concentrations. To statistically distinguish between nested models, the difference in the MOFV (−2 × log likelihood) was used because this difference is approximately χ^2^ distributed. A significance level of *p* < 0.005 equivalent to a difference in MOFV of 7.879 for 1 degree of freedom was considered. For non-hierarchical models, the most parsimonious model with the lowest objective function according to the Akaike information criterion (AIC) was considered [54].

#### 4.5.2. Covariate Model

Once the base model had been developed, the covariate analysis was carried out. Graphical exploration of potential correlations among continuous covariates and of individual PK parameters versus covariates was performed. The most physiologically and clinically relevant covariates were tested, on their respective PK parameters. Firstly, one covariate at a time was included, and then they were entered sequentially by the cumulative forward inclusion/backward elimination procedures. The inclusion of continuous covariates in their respective parameters was done in terms of exponential, linear or power relationships as appropriate. Covariates were centered on their median population value. Specifically, a power relationship was used to test body weight effect by either estimating the exponent or fixing it according to the allometry laws [55]. The statistical and clinical relevance of results of both approaches (estimated and fixed allometric exponents) were considered for model selection.

Significance levels of 5% (reduction in the MOFV of >3.841 units) and 0.1% (increase in the MOFV of >10.8 units) were applied during the forward addition and backward elimination steps. Other parameters considered for model selection were: precision expressed as relative standard error (RSE%), reductions in BPV associated with a specific PK parameter, model completion status (e.g., successful convergence or termination), η- and ε-shrinkage values [56], condition number estimated from the square root of the ratio of the major to the minor eigenvalue and visual inspection of goodness-of-fit plots with Xpose.

The following covariates were considered for screening: weight, BMI, gender, CL_CR_CKD-EPI, CL_CR_CG, CL_CR_MDRD-4, albumin concentration, neurocritical status, occurrence of drainage, occurrence of mechanical ventilation and treatment with vasoactive drugs.

#### 4.5.3. Model Evaluation

Throughout the modeling process, the following goodness-of-fit plots were examined: observed concentrations (DV) versus typical population model-predicted (PRED) or individual Bayesian-predicted concentrations (IPRED), conditional weighted residuals (CWRES) versus time and individual weighted residuals (IWRES) versus IPRED.

The stability and precision of the model were evaluated using a non-parametric bootstrap method [57]. Two hundred resamplings from the original dataset were performed. The median and 95% percentiles of the fixed and random effect parameters were calculated. The bias of each parameter was computed as the ratio of the difference between the median derived from the bootstrap and the final population estimate.

Prediction-corrected VPCs, based on 1000 replicated datasets from the original dataset, were constructed to prove the predictive capability of the model [58]. The 50%, 97.5th and 2.5th percentiles of the observations were checked to be within the non-parametric 95% confidence intervals for the 50%, 2.5th and 97.5th percentiles of the simulated profiles.

### 4.6. Monte Carlo Simulations

From the final model, Monte Carlo simulations were carried out to generate total concentrations-time profiles for 1000 subjects per dosing regimen. To achieve this, estimated fixed and random parameter values were fixed in the final model. Piperacillin dosage regimens of 8, 12, 16, 20 and 24 g daily given as CI up to the steady state and preceded by a loading dose of 4 g given in 30 min were evaluated. For each dosage, different CL_CR_CKD-EPI cut-offs, from 60 to 200 mL/min/1.73 m^2^ in steps of 20 mL/min/1.73 m^2^, were considered.

Considering 30% of protein binding [48], simulated total plasma concentrations were subsequently transformed into free concentrations (*f*C) by mathematical calculation as follows: *f*C = fu × C, where fu is the reported unbound fraction and C the measured piperacillin concentration.

### 4.7. Probability of Target Attainment and Cumulative Fraction of Response

From Monte Carlo simulations and for each dose and scenario, the PTA of two PK/PD targets, i.e., 100% *f*T_>1×MIC_ and 100% *f*T_>4×MIC_, were assessed. Calculations of PTA for pathogen MICs ranging from 1 to 16 mg/L were performed.

The CFR represents the probability of successful treatment by comparing the PTA with the MIC distribution of a specific population of microorganisms. The wild-type MIC distribution of *Pseudomonas aeruginosa* was obtained from the 2023 EUCAST database [59]. For each dosing regimen, CL_CR_CKD-EPI cut-off and PK/PD target, the CFR was calculated by multiplying the PTA found for each MIC by the proportion of isolates found at each MIC, as described previously [60]. Dosing was successful when the CFR was ≥90%.

## 5. Conclusions

The renal function estimated through CKD-EPI has been identified as the best predictor of inter-individual variability in piperacillin clearance. Our results suggest that in critically ill patients with preserved renal function, the actual dosage regimens have a risk of inadequate piperacillin exposure, which is aggravated with increasing renal function. For the PK/PD target of 100% *f*T_>1×MIC_, 12 g of piperacillin provide a PTA > 90% for MIC < 16 mg/L, regardless of CL_CR_, but higher doses are needed for MIC = 16 mg/L when CLCR > 100 mL/min/1.73 m^2^. For 100% *f*T_>4×MIC_, the highest dose (24 g/24 h) was not sufficient to ensure adequate exposure, except for MICs of 1 and 4 mg/L, regardless of CL_CR_. To reach the 100%*f*T_>4×MIC_ target in patients with CL_CR_ above 100 mL/min/1.73 m^2^, even the dose of 24 g CI would be insufficient to empirically treat susceptible *Pseudomonas aeruginosa* isolates (MIC ≤ 16 mg/L).

The developed model can be used as a support tool for dose guidance at the start of the treatment and during the TDM, based on renal function and the MIC of the causative pathogen.

## Figures and Tables

**Figure 1 antibiotics-12-00531-f001:**
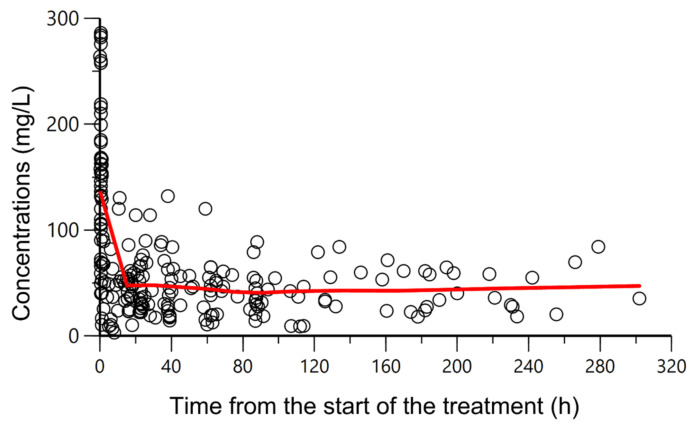
Overlapping observed piperacillin plasma concentrations (mg/L) (circles) versus time from the start of the treatment (h). A CI of piperacillin was administered, preceded by a loading dose of 4 g given over 30 min. Solid red line: smooth line indicating the general data trend.

**Figure 2 antibiotics-12-00531-f002:**
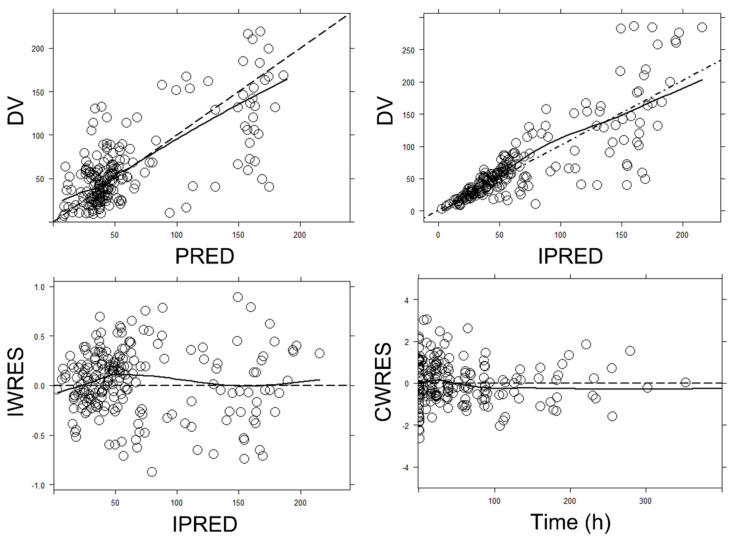
Goodness-of-fit plots for the final population pharmacokinetic model. **Upper left panel**: observed (DV) vs. population predicted concentrations (PRED). **Upper right panel**: DV vs. individual predicted concentrations (IPREDs). **Bottom left panel**: individual weighted residuals (IWRESs) vs. IPRED. **Bottom right panel**: population conditional weighted residuals (CWRESs) vs. time from the start of the treatment. Dashed line: identity line; Solid line: smooth line indicating the general data trend. Time was given in hours. Concentrations were given in mg/L.

**Figure 3 antibiotics-12-00531-f003:**
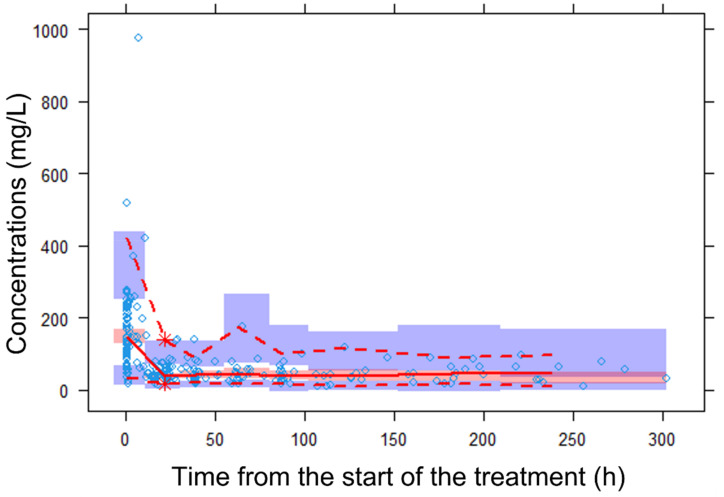
Prediction-corrected, visual predictive check (Predcorr-vpc) of the final pharmacokinetic model for piperacillin. The diamond represents the observed data. The red dashed lines depict the 2.5th and 97.5th percentiles of the observed concentrations. The solid line corresponds to the 50th percentiles of the observed concentrations. The red and blue bands represent the 95% prediction intervals of the 50% and 2.5% and 97.5% percentiles of the simulated data. Predcorr-vpc showed that most of the observed concentrations fell within the 90% prediction interval of the simulated data and were randomly distributed around the median.

**Figure 4 antibiotics-12-00531-f004:**
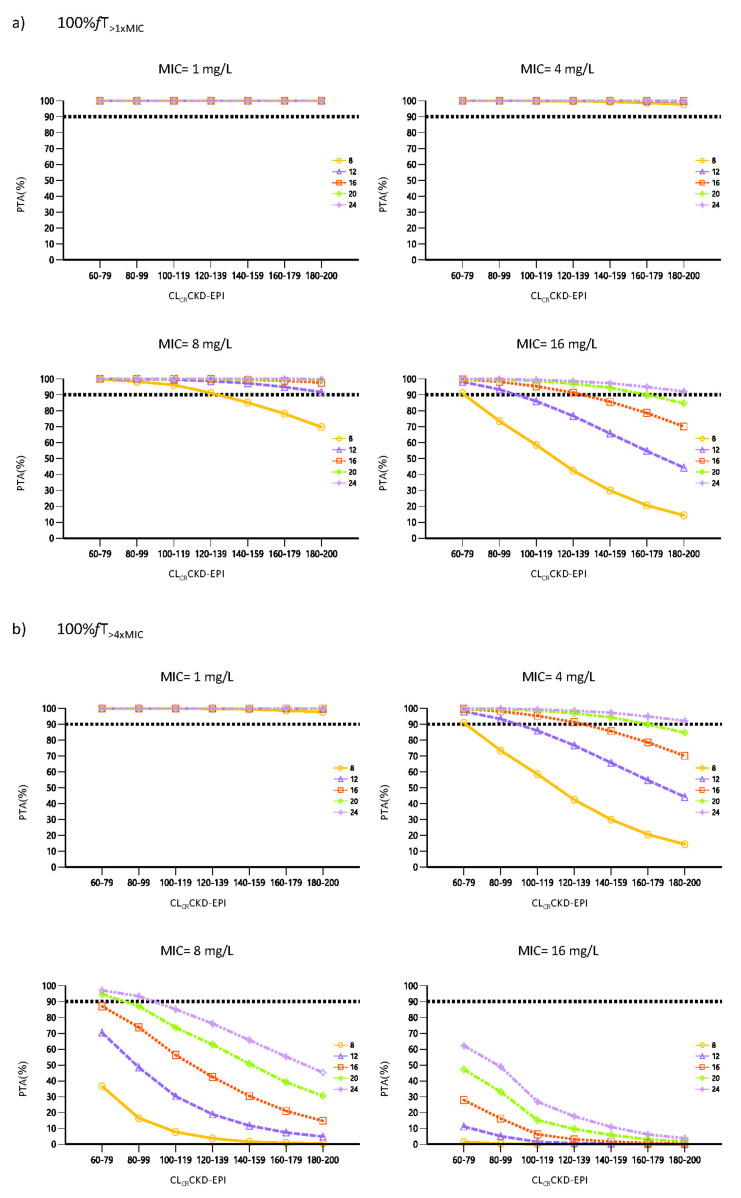
Probability of Target Attainment (PTA) for free piperacillin plasma concentrations at steady state. The prediction shows the PTA at 5 daily doses levels of 8, 12, 16, 20 and 24 g/24 h given as CI varying renal clearance (cut-offs ranging from 60–79 to 180–200 mL/min/1.73 m^2^). The MIC values of 1, 4, 8 and 16 mg/L were evaluated considering 2 different PK/PD targets (**a**) 100% *f*T_>1×MIC_, (**b**) 100% *f*T_>4×MIC_. The horizontal dashed line indicates the 90% of PTA.

**Table 1 antibiotics-12-00531-t001:** Demographic and baseline clinical characteristics.

Variable	Value
Number of patients, n	106
Age (years)	65 (50–72)
Males, n (%)	71 (66.98%)
Weight (kg)	72 (65–84)
BMI (kg/m^2^)	26.2 (23–29)
Caucasian, n (%) CL_CR_CKD-EPI (mL/min/1.73 m^2^)	102 (96.23%) 97.1 (86–114)
CL_CR_CG (mL/min) CL_CR_MDRD-4 (mL/min/1.73 m^2^) Albumin (g/L) Neurocritical status, n (%) Drainage carriers, n (%) Ventilated, n (%) Vasoactive drugs, n (%)	109.4 (84–143) 116.3 (89–146) 29 (26–33) 36 (33.96%) 37 (34.91%) 63 (59.43%) 45 (42.45%)

BMI, Body Mass Index; CL_CR_, creatinine clearance; CKD-EPI, Chronic Kidney Disease Epidemiology Collaboration; CG, Cockcroft and Gault; MDRD-4, Modification of Diet in Renal Disease. Continuous variables are presented as median (interquartile range) and dichotomous variables as n (%).

**Table 2 antibiotics-12-00531-t002:** Base model, final model and bootstrap parameter estimates and variabilities from the population pharmacokinetic modeling analysis, including uncertainty and shrinkage.

Parameter	Base Model Parameter Estimates (*RSE*%) [SHR%]	Final Model Parameter Estimates (*RSE*%) [SHR%]	Median (95% CI) Bootstrap Results ^a^
Pharmacokinetic parameter			
CL (L/h)	11.9 (*7.19*)	12.0 (*6.03*) × (CL_CR_CKD-EPI/99.24)	11.95 (10.36–13.42)
Vc (L)	20.4 (*10.00*)	20.7 (*8.94*)	20.29 (9.20–25.05)
Vp (L)	65.3 (*58.04*)	62.4 (*50.80*)	66.32 (15.45–181.76)
CL_D_ (L/h)	5.6 (*30.00*)	4.77 (*26.4*)	5.16 (3.19–52.05)
Between-patient variability			
ω^2^_CL_	0.253 (*22.41*) [14.60]	0.190 (*24.42*) [17.16]	0.181 (0.103–0.296)
Residual variability			
σ^2^	0.148 (*15.27*) [14.11]	0.140 (*15.7*) [13.45]	0.135 (0.099–0.181)

CL, plasma clearance; V_C_, central compartment distribution volume; V_P_, peripheral compartment distribution volume; CL_D_, distributional clearance between central and peripheral compartments; ω^2^_CL_, variance of between-patient random effects with CL; σ^2^, variance of residual random effects; RSE, relative standard error; SHR, shrinkage. ^a^ Derived from 200 successful bootstrap re-samplings. In the final model, the relationship between the typical clearance value (TVCL) and the CL_CR_CKD-EPI was given by the equation: TVCL= θ × (CL_CR_CKD-EPI/99.24), where θ refers to the piperacillin clearance for the typical patient of CL_CR_CKD-EPI = 99.24 mL/min/1.73 m^2^.

**Table 3 antibiotics-12-00531-t003:** Cumulative fraction of response for piperacillin over ranges of creatinine clearance values.

PK/PD Target	CL_CR_CKD-EPI (mL/min/1.73 m^2^)	CFR (%) According to Different Daily Dose of Piperacillin in CI (g)				
		8	12	16	20	24
100% *f*T_>1×MIC_	60–79	98.64	99.74	99.97	99.98	100.00
	80–99	95.77	99.02	99.74	99.93	99.98
	100–119	93.11	97.89	99.33	99.82	99.91
	120–139	89.52	96.33	98.74	99.55	99.79
	140–159	86.10	94.42	97.83	99.17	99.61
	160–179	82.67	92.22	96.65	98.48	99.26
	180–200	79.28	89.93	95.15	97.68	98.83
100% *f*T_>4×MIC_	60–79	65.80	79.23	86.45	91.10	93.87
	80–99	51.08	70.50	80.77	87.08	91.01
	100–119	40.53	61.55	73.64	80.87	85.64
	120–139	30.46	53.51	67.69	76.57	81.73
	140–159	22.88	45.57	61.41	71.78	77.61
	160–179	17.39	38.30	55.09	66.08	73.15
	180–200	13.74	31.77	48.76	60.92	68.87

PK/PD target, pharmacokinetic/pharmacodynamic target; 100% *f*T_>1×MIC_, 100% time above the minimum inhibitory concentration; 100% *f*T_>4×MIC_, 100% time above 4 times the minimum inhibitory concentration; CL_CR_CKD-EPI, creatinine clearance estimated by the Chronic Kidney Disease Epidemiology Collaboration formula; CFR, cumulative fraction of response; CI, continuous infusion. CFR for susceptible *Pseudomonas aeruginosa* isolates (MIC ≤ 16 mg/L) at different CI regimens (preceded by a 4 g piperacillin loading dose) and different PK/PD targets over different cut-offs of CL_CR_CKD-EPI.

## Data Availability

Main data will be made available on request to the corresponding author.

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
