# Peer review of "Predictive Factors of Piperacillin Exposure and the Impact on Target Attainment after Continuous Infusion Administration to Critically Ill Patients"

_antibiotics, 2023, doi:10.3390/antibiotics12030531_

Round 1
Reviewer 1 Report
The title of the paper is about piperacillin but patients received piperacillin/tazobactam. Data are only about piperacillin, but since tazobactam should have profound effects on piperacillin activity and target attainment, the study should take into account the effect of the beta-lactamse inhibitor, or at least try to evaluate it in some way in discussion.
The conclusion about inadequate target attainment, in the absence of an evaluation of tazobactam effect seems inconclusive.
Author Response
Response to Reviewer 1 Comments
Point 1: The title of the paper is about piperacillin but patients received piperacillin/tazobactam. Data are only about piperacillin, but since tazobactam should have profound effects on piperacillin activity and target attainment, the study should take into account the effect of the beta-lactamse inhibitor, or at least try to evaluate it in some way in discussion.
The conclusion about inadequate target attainment, in the absence of an evaluation of tazobactam effect seems inconclusive.
Response 1:
As indicated in the manuscript, we are aware that the lack of measurement of tazobactam serum concentrations is one of the limitations of the current study. Tazobactam is a beta lactamase inhibitor that prevents piperacillin hydrolysis and contributes to enhance its exposure and, thus, its efficacy. There is enough evidence supporting the existence of a high correlation between piperacillin and tazobactam concentrations (1–4). Previous literature checked during the study design, showed that no PK interaction occurs between piperacillin and tazobactam (5,6). Tazobactam concentrations have been assessed in few population pharmacokinetic studies, in which results showed that concentrations were above the threshold at almost all times, so that the efficiency was considered to depend on piperacillin concentrations(1,2,5,6). The discussion has been modified to further develop this limitation.
“Patients were treated with P/T but only piperacillin concentrations were measured. There is enough evidence supporting the existence of a high correlation between piperacillin and tazobactam concentrations (1–4). In fact, a recent PK study showed that tazobactam concentrations were above the threshold in 92% of cases in which the piperacillin PK/PD target was achieved (4). The similarity in the PK of these two compounds means that the determination of tazobactam is often considered dispensable and, consequently, studies in which it is evaluated are scarce. In previous PPK studies carried out in patients undergoing RRT (5,6), simulated tazobactam concentrations were found to be above threshold at all times. In a more recent study in which P/T was administered by II, tazobactam concentrations were predicted in critically ill patients with creatinine clearance up to 200 mL/min (1,2). The tazobactam target was reached in 94% and 99% of the patients when doses of 1.5g/24h and ≥ 2g/24h were administered, thus P/T efficacy was considered to be uniquely related to piperacillin concentration. Although Wallemburg et al. considered the development of a joint model to adequately predict piperacillin concentrations (1,2), previous studies showed there was no interaction between and the PK/PD analysis was therefore based solely on the simulations performed with the piperacillin model (5,6). Additional studies assessing tazobactam concentrations are needed to confirm the existence of such an association.”
- Wallenburg E, ter Heine R, Schouten JA, Raaijmakers J, ten Oever J, Kolwijck E, et al. An Integral Pharmacokinetic Analysis of Piperacillin and Tazobactam in Plasma and Urine in Critically Ill Patients. Clin Pharmacokinet [Internet]. 2022;61(6):907–18. Available from: https://doi.org/10.1007/s40262-022-01113-6
- Wallenburg E, ter Heine R, Schouten JA, Raaijmakers J, ten Oever J, Kolwijck E, et al. Correction to: An Integral Pharmacokinetic Analysis of Piperacillin and Tazobactam in Plasma and Urine in Critically Ill Patients (Clinical Pharmacokinetics, (2022), 61, 6, (907-918), 10.1007/s40262-022-01113-6). Clin Pharmacokinet [Internet]. 2022;61(9):1325–9. Available from: https://doi.org/10.1007/s40262-022-01165-8
- Occhipinti DJ, Pendland SL, Schoonover LL, Rypins EB, Danziger LH, Rodvold KA. Pharmacokinetics and Pharmacodynamics of Two Multiple-Dose Piperacillin-Tazobactam Regimens. 1997;41(11):2511–7.
- Zander J, Döbbeler G, Nagel D, Scharf C, Huseyn-Zada M, Jung J, et al. Variability of piperacillin concentrations in relation to tazobactam concentrations in critically ill patients. Int J Antimicrob Agents [Internet]. 2016;48(4):435–9. Available from: http://dx.doi.org/10.1016/j.ijantimicag.2016.06.013
- Asín-Prieto E, Rodríguez-Gascón A, Trocóniz IF, Soraluce A, Maynar J, Sánchez-Izquierdo JÁ, et al. Population pharmacokinetics of piperacillin and tazobactam in critically ill patients undergoing continuous renal replacement therapy: Application to pharmacokinetic/pharmacodynamic analysis. J Antimicrob Chemother. 2014;69(1):180–9.
- Tamme K, Oselin K, Kipper K, Tasa T, Metsvaht T, Karjagin J, et al. Pharmacokinetics and pharmacodynamics of piperacillin/tazobactam during high volume haemodiafiltration in patients with septic shock. Acta Anaesthesiol Scand. 2016;60(2):230–40.
Reviewer 2 Report
The manuscript “Predictive factors of piperacillin exposure and impact on target attainment, when given as continuous infusion in critically ill patients” was revised. In general, the paper is very interesting, well written, with novel results for the research field. I have few observations that can be adjusted before acceptance:
1) Figure 4 can be improved. It is very difficult to understand the information.
2) Table 3. I suggest avoiding using colors in tables. If authors want to express different, I suggest including a symbol.
3) The conclusion can be improved. Please provide some numeric results as well as the future perspectives and real applicability of these results in the research field.
Author Response
Response to Reviewer 2 Comments
Point 1: Figure 4 can be improved. It is very difficult to understand the information.
Response 1: The figure has been edited to make it easier to be interpreted.
Point 2: Table 3. I suggest avoiding using colors in tables. If authors want to express different, I suggest including a symbol.
Response 2: Shaded areas have been removed from the table.
Point 3: The conclusion can be improved. Please provide some numeric results as well as the future perspectives and real applicability of these results in the research field.
Response 3: Conclusions have been modified and expanded as requested:
“The renal function estimated through CKD-EPI has been identified as the best predictor of inter-individual variability in piperacillin clearance. Our results suggest that in critically ill patients with preserved renal function, the actual dosage regimens have a risk of inadequate piperacillin exposure, which is aggravated with increasing renal function. For the PK/PD target of 100%fT>1xMIC, 12g of piperacillin provide a PTA>90% for MIC<16mg/L regardless of CLCR but higher doses are needed for MIC=16 mg/L when CLCR>100 mL/min/1.73m2. For 100%fT>4xMIC, the highest dose (24g/24h) was not sufficient to ensure adequate exposure, excepting for MICs of 1 and 4 mg/L regardless of CLCR. To reach the 100%fT>4xMIC target in patients with CLCR above 100 ml/min/1.73m2, even the dose of 24 g/24h CI would be insufficient to empirically treat susceptible Pseudomonas aeruginosa isolates (MIC≤16 mg/L).
The developed model can be used as a support tool for dose guidance at the start of the treatment and during the TDM, based on renal function and the MIC of the causative pathogen.”
Round 2
Reviewer 1 Report
Authors properly discussed our previous concerns and provided bibliographic support for not having measured tazobactam concentrations